# Modified Fe_3_O_4_ Nanoparticles for Foam Stabilization: Mechanisms and Applications for Enhanced Oil Recovery

**DOI:** 10.3390/nano15050395

**Published:** 2025-03-04

**Authors:** Dandan Yin, Judong Qiu, Dongfeng Zhao, Yongzheng Wang, Tao Huang, Yunqian Long, Xiaohe Huang

**Affiliations:** 1College of Petrochemical Engineering & Environment, Zhejiang Ocean University, Zhoushan 316021, China; aayindan@163.com (D.Y.); qiujudong@163.com (J.Q.); huangtaozjou@126.com (T.H.); longyunqian@zjou.edu.cn (Y.L.); hxh_0258@163.com (X.H.); 2Zhejiang Key Laboratory of Pollution Control for Port-Petrochemical Industry, Zhoushan 316022, China

**Keywords:** Fe_3_O_4_ nanoparticles, foam stabilization, enhanced oil recovery, magnetic nanoparticles, hydrophobic modification

## Abstract

Nanoparticles (NPs) have shown great potential in stabilizing foam for enhanced oil recovery (EOR). However, conventional NPs are difficult to recover and may contaminate produced oil, increasing operational costs. In contrast, superparamagnetic Fe_3_O_4_ NPs can be efficiently recovered using external magnetic fields, offering a sustainable solution for foam stabilization. In this study, Fe_3_O_4_ NPs were coated with SiO_2_ using tetraethyl orthosilicate (TEOS) and further modified with dodecyltrimethoxysilane to enhance their hydrophobicity. The modification effects were characterized, and the optimal foam-stabilizing Fe_3_O_4_@SiO_2_ NPs were found to have a contact angle of 77.01°. The foam system formed with α-olefin sulfonate (0.2 wt%) as the foaming agent and the optimal modified NPs exhibited a drainage half-life of 452 s. After foam-stabilization experiments, the NPs were recovered and reused, with the results indicating that three recovery cycles were optimal. Finally, visual microscopic displacement experiments demonstrated that the foam stabilized by modified NPs effectively mobilized clustered, membranous, and dead-end residual oil, increasing the recovery rate by 17.01% compared with unmodified NPs. This study identifies key areas for future investigation into the application of magnetic nanoparticles for enhanced oil recovery.

## 1. Introduction

In late-stage reservoir development, reservoir heterogeneity intensifies and the distribution of remaining oil becomes increasingly scattered, leading to a significant decline in oil recovery [1]. Foam flooding has been proposed as an effective method to enhance oil recovery by blocking high-permeability zones and increasing the viscosity of the displacing fluid, thereby improving sweep efficiency [2,3]. However, the thermodynamic instability of foam severely limits its industrial application [4,5].

To enhance foam stability, surfactants are commonly used to reduce surface tension, while polymers can enhance the mechanical strength of the liquid film [6,7,8,9]. Although these methods can temporarily stabilize foam, the degradation and loss of surfactants and polymers at high temperatures in the reservoir make it difficult to maintain the foam’s stability for a long time [10]. Nanoparticles (NPs) can enhance foam stability by adsorbing at the bubble surface, preventing coalescence and coarsening, and by reinforcing the liquid film, delaying foam drainage [11,12,13,14]. Moreover, NPs maintain stability in high-temperature and high-salinity reservoirs [15,16,17,18,19]. However, conventional NPs cannot be recovered after use, increasing operational costs and complicating the treatment of produced oil.

Fe_3_O_4_ nanoparticles, with their superparamagnetic properties, can rapidly respond to external magnetic fields, enabling efficient recovery and reuse [20,21], which significantly reduces costs and environmental impact. Nevertheless, unmodified Fe_3_O_4_ NPs tend to aggregate and exhibit poor oxidation stability, limiting their effectiveness in foam stabilization. To address these limitations, surface modification has been researched. Research on the modification of Fe_3_O_4_ NPs primarily focuses on improving their dispersibility, stability, interfacial adsorption properties [22], and adaptability to complex reservoir conditions [23]. Common modification methods include surface coating (e.g., SiO_2_, polymers) [24], chemical modification (e.g., silane coupling agents), and functionalization (e.g., hydrophobic modification) [25,26]. SiO_2_ coating significantly enhances the oxidation resistance and dispersibility of Fe_3_O_4_ NPs while improving their stability under high-temperature and high-salinity conditions [18,27]. Hydrophobic modification using silane coupling agents (e.g., WD-10) can regulate the wettability of Fe_3_O_4_ NPs [28,29], enabling them to adsorb at the gas–liquid interface and form a physical barrier that prevents bubble coalescence and coarsening [30,31], while also strengthening the liquid film and delaying foam drainage, thereby enhancing foam stability. Additionally, polymer modification (e.g., polyethylene glycol) can improve the biocompatibility and environmental friendliness of Fe_3_O_4_ NPs, reducing their environmental impact in oilfield applications [32,33].

Despite the progress in foam-stabilization technology based on Fe_3_O_4_ NPs, challenges such as particle aggregation and insufficient surface modification remain [34]. Further optimization through shell design and functional modification is required. Moreover, most studies have focused on macroscopic foam stability and oil displacement efficiency, failing to reveal the processes of foam generation, migration, and residual oil mobilization using modified Fe_3_O_4_ NPs [35]. To address these issues, this study focuses on the modification of Fe_3_O_4_ NPs through silica coating and hydrophobic functionalization, aiming to enhance their foam-stabilization performance and recyclability. The modified NPs are characterized, and their foam-stabilization mechanisms are investigated through static foam stability tests and microscopic visualization experiments. Additionally, the recyclability of the modified NPs is evaluated, and their performance in mobilizing clustered, film, and dead-end residual oil is systematically studied using a microfluidic model that mimics real reservoir conditions.

The novelty of this work lies in its comprehensive investigation of the foam-stabilization and enhanced oil recovery mechanisms of modified Fe_3_O_4_ NPs, particularly their ability to mobilize different types of residual oil in complex pore networks. By combining macroscopic foam stability tests with microscopic visualization experiments, this study provides new insights into the pore-scale mechanisms of NP-stabilized foam flooding, offering valuable theoretical and technical support for the application of magnetic NPs in enhanced oil recovery.

## 2. Materials and Methods

### 2.1. Materials

This experimental research used Fe_3_O_4_ nanoparticles (NPs) with an average particle size of 20 nm, purchased from Macklin Biochemical Co., Ltd. (Shanghai, China). Additionally, hydrochloric acid (36.0–38.0%), absolute ethanol (99.7%), and ammonia solution (25.0–28.0%) were obtained from Sinopharm Chemical Reagent Co., Ltd. (Shanghai, China). Other chemicals included tetraethyl orthosilicate (TEOS, 99%), n-hexane (97%), KH-304 (C₁₅H_34_O_3_Si, 97%), and α-olefin sulfonate (AOS, 92%), all sourced from Macklin Biochemical Co., Ltd. All reagents were of analytical grade and used without further purification.

### 2.2. Methods

#### 2.2.1. Modification of Fe_3_O_4_ NPs

To mitigate the mutual attraction between Fe_3_O_4_ NPs, a silica layer was deposited onto the Fe_3_O_4_ NPs using tetraethyl orthosilicate (TEOS) as the precursor. The synthesis of Fe_3_O_4_@SiO_2_ NPs was carried out via a modified Stöber method based on the sol–gel process, as illustrated in Figure 1. Specifically, 1 g of Fe_3_O_4_ NPs was dispersed in 50 mL of 0.1 mol/L HCl solution under ultrasonic (pulse mode: 5 s on/2 s off) treatment for 10 min to activate the hydroxyl groups on the surface of NPs, thereby facilitating the subsequent SiO_2_ coating. The Fe_3_O_4_ NPs were then washed five times with deionized water and ethanol to remove impurities, followed by dispersion in 80 mL of an ethanol solution (with a volume fraction of 80%). After ultrasonic dispersion for 10 min, the mixture was transferred to a stirring apparatus and stirred at 720 r/min for 20 min. Subsequently, 12 mL of NH_3_·H_2_O (with a concentration of 25~28%) was added, and the stirring was continued for an additional 30 min. Following this, varying volumes of tetraethyl orthosilicate (TEOS) (800 μL, 1000 μL, and 1200 μL) were introduced into C_2_H_5_OH/H_2_O solution (with a volume fraction of 80%) and stirred at room temperature for 5 h. The resulting solution was centrifuged, washed repeatedly with ethanol and deionized water (3–4 times), and dried at 70 °C for 12 h to obtain the Fe_3_O_4_@SiO_2_ NPs.

The synthesized Fe_3_O_4_@SiO_2_ nanoparticles (NPs) were further functionalized using dodecyltrimethoxysilane (KH-304), as depicted in Figure 2. Specifically, a measured quantity of KH-304 was introduced into 50 mL of 90% ethanol solution and subjected to hydrolysis under continuous stirring at 75 °C for 20 min, resulting in the formation of solution D. Concurrently, 0.2 g of Fe_3_O_4_@SiO_2_ NPs was dispersed in 50 mL of 90% ethanol solution via ultrasonication for 10 min to obtain solution E. Solutions D and E were then combined and stirred at 75 °C with a stirring speed of 400–450 rpm for 5 h. Following the reaction, the mixture was centrifuged, washed 3–4 times with ethanol and deionized water, and dried at 70 °C for 6 h to yield the surface-modified Fe_3_O_4_@SiO_2_ NPs.

#### 2.2.2. Characterization Techniques

The morphological characteristics of the NPs were examined using scanning electron microscopy (SEM; ZEISS SUPRA-55, Oberkochen, Germany), and the particle size distribution was quantitatively assessed using Image-J software (V1.8.0.112). Surface functional groups of the NPs were analyzed by Fourier-transform infrared spectroscopy (FTIR; Nicolet 6700, Thermo Scientific, Waltham, MA, USA) employing the potassium bromide (KBr) pellet method. Additionally, the hydrophobicity of the modified NPs was evaluated by measuring the contact angle using a contact angle goniometer (JY-82A).

#### 2.2.3. Evaluation of Foam Stabilization and NPs Recovery

The foam stability was assessed using the Waring blender method [36]. Specifically, Fe_3_O_4_@SiO_2_ NPs with different wettability modifications (0.2 wt%) were dispersed in 200 mL of a 0.2 wt% AOS solution using an ultrasonic disperser to prepare a homogeneous Fe_3_O_4_@SiO_2_ (0.2 wt%)-AOS (0.2 wt%) dispersion. The dispersion was then foamed using a high-speed blender (8011ES, Waring, Stamford, CT, USA) at 5000 rpm for 3 min, and the resulting foam was transferred to a graduated cylinder. Foamability was characterized by measuring the initial foam volume, while foam stability was evaluated based on the drainage half-life, defined as the time required for half of the liquid to drain. The NPs exhibiting the optimal contact angle were selected for further foam-stabilization experiments. After foam collapse, the mixed solution was subjected to an external magnetic field to facilitate NP sedimentation. The recovered NPs were washed with distilled water and reused for subsequent foam-stabilization tests. The changes in foam half-life were recorded to evaluate the reusability and performance of NPs.

#### 2.2.4. Adsorption of NPs on Liquid Films

The distribution of NPs within the foam system was investigated using a fluorescence microscope (Trim Scope, Wilnsdorf, Germany) to elucidate the stabilization mechanism of Fe_3_O_4_@SiO_2_ NPs in foam. For this purpose, the fluorescent probe rhodamine B, which carries a negative charge and exhibits a maximum excitation wavelength of 543 nm, was employed to label the NPs in the dispersion. The stained NPs were subsequently centrifuged and washed repeatedly with distilled water until the supernatant became clear. The labeled NPs were then utilized to generate foam, and fluorescence images of the foam were captured using the microscope to analyze the spatial distribution and behavior of NPs within the foam structure.

#### 2.2.5. Microscopic Oil Displacement Experiment

Microscopic oil displacement experiments were conducted using a micro-etched porous media model designed to replicate the natural pore structure of cores from the Daqing oilfield. The pore network was etched onto a glass plate using photolithography, and the etched plate was bonded to a smooth glass plate to create the microscopic model. The fabricated glass micromodel exhibited a porosity of 25%, with an average pore depth of approximately 40 μm and a pore width of approximately 100 μm. The experimental setup, as illustrated in Figure 3, comprised a foam generation device, a microscopic pore model, and an observation data acquisition system.

Prior to the experiment, the micromodel was evacuated and saturated with crude oil (viscosity = 7.69 mPa·s at 25 °C). Water flooding was initiated by injecting 5 pore volumes (PVs) of water until water without oil was produced. Foam was generated by co-injecting a foam solution and air at a 1:1 volume ratio into the foam generator. Subsequently, 2.5 PVs of either AOS foam or Fe_3_O_4_@SiO_2_-AOS-stabilized foam were injected, followed by 2.5 PVs of water flooding at a constant injection rate of 1 μL/min. The entire displacement process was monitored and recorded using a microscope (VHX-5000, Keyence, Osaka, Japan) equipped with a high-resolution CCD camera and a parallel light source. The experiments focused on investigating the mobilization of various types of residual oil by Fe_3_O_4_@SiO_2_-stabilized foam under high water-saturation conditions at 25 °C.

After the experiment, the micromodel was cleaned by injecting petroleum ether and ethanol to remove residual oil and ensure all pores and throats were free of contaminants. The micromodel was then dried in a constant temperature oven at 45 °C for subsequent use.

## 3. Results

### 3.1. Characterization of NPs

Following the modification process, a portion of the hydroxyl (-OH) groups on the surface of the NPs were substituted with carbon chains, resulting in a reduction in saturation magnetization. The successful formation of the modified NPs was confirmed through comprehensive characterization using scanning electron microscopy (SEM), transmission electron microscopy (TEM), and Fourier-transform infrared spectroscopy (FTIR). These techniques were employed to evaluate the morphological and structural changes, as well as to analyze the surface functional groups of the modified NPs.

#### 3.1.1. Morphological Analysis

The SEM images of Fe_3_O_4_@SiO_2_ NPs synthesized with varying amounts of TEOS (800 μL, 1000 μL, and 1200 μL) are presented in Figure 4, illustrating the morphological and size characteristics of the NPs. Following the SiO_2_ coating process, the particle size increased significantly from 20 nm to approximately 300 nm. The SEM analysis also revealed the enhanced dispersibility of NPs after coating. As depicted in Figure 4a, bare Fe_3_O_4_ NPs displayed irregular shapes and pronounced agglomeration. In contrast, Figure 4b shows Fe_3_O_4_@SiO_2_-800 NPs, which exhibited a flocculent structure with reduced agglomeration, suggesting that the SiO_2_ coating mitigated the interparticle interactions of Fe_3_O_4_ NPs. However, due to the insufficient amount of TEOS, a well-defined spherical core–shell structure was not achieved. Figure 4c,d display Fe_3_O_4_@SiO_2_-1000 NPs and Fe_3_O_4_@SiO_2_-1200 NPs, respectively. These NPs demonstrated spherical morphologies, significantly larger particle sizes, reduced agglomeration, and more uniform structures, confirming the successful formation of an SiO_2_ coating on the Fe_3_O_4_ NPs’ surface.

To evaluate the size distribution of Fe_3_O_4_@SiO_2_-1000 NPs and Fe_3_O_4_@SiO_2_-1200 NPs, particle size analysis was conducted using Image-J software by randomly selecting 80 points from each SEM image. The results are presented in Figure 5. As depicted in Figure 5a, the particle size distribution of bare Fe_3_O_4_ NPs ranges from 20 to 200 nm, with an average diameter of 57.663 nm. Figure 5b illustrates that the particle size distribution of Fe_3_O_4_@SiO_2_-1000 NPs spans from 60 to 240 nm, with an average diameter of 147.252 nm. Similarly, Figure 5c demonstrates that the particle size distribution of Fe_3_O_4_@SiO_2_-1200 NPs extends from 60 to 320 nm, with an average diameter of 158.021 nm. These findings indicate a significant increase in the diameter of the NPs following the core–shell coating process, with the particle size positively correlated with the amount of TEOS used. Furthermore, compared with Fe_3_O_4_@SiO_2_-1200 NPs, Fe_3_O_4_@SiO_2_-1000 NPs exhibit a narrower particle size distribution, a smaller average particle size, and enhanced uniformity. Consequently, 1000 μL of TEOS was determined to be the optimal amount for coating the Fe_3_O_4_ NPs.

To visually confirm the formation of the core–shell structure, the NPs were characterized using TEM and Energy-Dispersive X-ray Spectroscopy (EDS), as illustrated in Figure 6. In Figure 6a, the bare Fe_3_O_4_ NPs exhibit a uniform granular morphology, consistent with the observations from SEM imaging. Figure 6b reveals the core–shell structure of Fe_3_O_4_@SiO_2_-1000 NPs, where the core (Fe_3_O_4_) appears darker, smaller, and denser, while the shell (SiO_2_) appears lighter, larger, and forms a well-defined spherical coating. This confirms the successful deposition of a silica shell on the surface of the Fe_3_O_4_ NPs.

To further validate the chemical composition of the materials, EDS analysis was performed. The results indicate that the bare Fe_3_O_4_ NPs consist of 64.6% Fe and 35.4% O, whereas the SiO_2_-coated Fe_3_O_4_ NPs comprise 48.8% O, 33.0% Fe, and 18.2% Si. The presence of a weak Si peak alongside a strong Fe peak provides additional evidence for the successful formation of a silica shell on the surface of the Fe_3_O_4_ NPs. These findings collectively demonstrate the effective synthesis of the core–shell structure.

#### 3.1.2. FTIR Analysis

FTIR analysis was employed to confirm the presence of grafted functional groups and the successful formation of modified NPs, as depicted in Figure 7. In all four spectra, the characteristic Fe-O bond vibration peak was observed at 586 cm^−1^ and 630 cm^−1^, confirming the presence of Fe_3_O_4_. Additionally, peaks corresponding to -OH stretching and bending vibrations were identified at 3310 cm^−1^ and 1627 cm^−1^, respectively [37]. A strong absorption peak at 1082 cm^−1^ was attributed to the asymmetric stretching vibration of the Si-O-Si bond, while the peak at 462 cm^−1^ represented the bending vibration of the Si-O-Si bond, both of which are indicative of the SiO_2_ coating. The peak at 1123 cm^−1^, corresponding to the Fe-O-Si stretching vibration, further confirmed the successful formation of the SiO_2_ coating and the core–shell structure. In the FTIR spectrum of Fe_3_O_4_@SiO_2_+KH-304, the peaks observed at 2924 cm^−1^ and 2831 cm^−1^ were assigned to the asymmetric stretching vibrations of the -CH_3_ and -CH_2_ groups, respectively, indicating the presence of carbon chains on the NP surface [38]. Since these carbon chains originate from the hydrophobic modifier KH-304, these results provide clear evidence of the successful hydrophobic modification of the NPs.

### 3.2. Analysis of Surface Hydrophobicity and Foam-Stabilization Ability of Modified NPs

#### 3.2.1. Analysis of Surface Hydrophobicity of Modified NPs

NPs must exhibit moderate hydrophobicity to ensure strong adsorption at the gas–liquid interface. However, excessively hydrophobic NPs are ineffective in stabilizing foam, as they fail to adsorb onto the lamellae and prevent liquid drainage [39]. Therefore, it is crucial to select NPs with an optimal level of hydrophobicity.

The contact angles of Fe_3_O_4_@SiO_2_-1000-KH-304 NPs, modified with varying amounts of KH-304, were measured, as illustrated in Figure 8. As the concentration of KH-304 increased, the contact angle between the NPs and water progressively rose from 57.18° to 121.34°, after which it plateaued, indicating that further increases in the modifier did not alter the contact angle. To investigate the influence of NPs with different wettabilities on foam stability, NPs with contact angles of 57.18°, 77.01°, 105.51°, and 121.34° were selected for subsequent foam stability experiments.

#### 3.2.2. Analysis of Foam-Stabilization Ability of Modified NPs

Foam was generated by high-speed stirring of a foaming solution containing Fe_3_O_4_@SiO_2_-1000-KH-304 NPs with varying hydrophobicities and alpha-olefin sulfonate (AOS) at a concentration of 0.2 wt%. The drainage half-life and initial foam volume were measured and are presented in Figure 9. The results demonstrated that the foam-stabilization effect was optimal at an NP concentration of 1.0 wt% and a contact angle of 77.01°, achieving a drainage half-life of 452 s and an initial foam volume of 664 mL. In contrast, at the same NP concentration of 1.0 wt%, the foam-stabilization capability was weakest for NPs with a contact angle of 123.3°, exhibiting a drainage half-life of only 374 s, which was lower than that of unmodified NPs.

#### 3.2.3. Adsorption of NPs on Foam Surfaces

To elucidate the mechanism by which Fe_3_O_4_@SiO_2_ NPs stabilize foam, the NPs were labeled with a fluorescent dye, and their distribution within the foam system was observed using fluorescence microscopy, as illustrated in Figure 10. Figure 10a reveals that the NPs were predominantly distributed in the liquid phase between bubbles. To more clearly visualize the particle distribution, the foam was drained to form dry foam. In the Fe_3_O_4_@SiO_2_ NPs-AOS (0.2 wt%) system, significant NP loss was observed, with few NPs adsorbed on the liquid film, as shown in Figure 10b. In contrast, the majority of Fe_3_O_4_@SiO_2_ NPs with a contact angle of 77.01° were adsorbed onto the liquid film of the bubbles and remained in place even after liquid drainage, as depicted in Figure 10c. Conversely, Fe_3_O_4_@SiO_2_ NPs with a contact angle of 121.34° tended to migrate out of the liquid phase and into the gas phase, with minimal adsorption on the liquid film, as shown in Figure 10d.

These observations indicate that NPs located in the surrounding continuous liquid phase are carried away during liquid drainage, whereas NPs adsorbed at the gas–liquid interface remain within the dry foam skeleton after drainage. Specifically, NPs with a contact angle of 77.01° adsorb onto the bubble surface, forming a protective particle armor. Unmodified core–shell NPs, being more hydrophilic, predominantly reside in the liquid phase, while NPs with a contact angle of 121.34°, due to their stronger hydrophobicity, preferentially remain in the gas phase. These findings highlight the critical role of NPs’ hydrophobicity in determining their distribution and foam-stabilizing behavior.

### 3.3. Recyclability of NPs

Fe_3_O_4_ NPs exhibit a high responsiveness to external magnetic fields. Consequently, after assessing foam stability, the NPs were recovered using an external magnetic field and reused for foam stabilization. The foam solution employed for recyclability evaluation consisted of 1 wt% Fe_3_O_4_@SiO_2_ NPs (77.01°) and 0.2 wt% AOS, representing the optimal system identified earlier. Following foam generation, the foam half-life was recorded. The NPs were then recovered by applying a magnetic field, washed multiple times with ethanol and deionized water, and reused for foam stabilization until a significant reduction in drainage half-life was observed.

As shown in Figure 11a, the mixed solution exhibited a noticeable lightening in color after 5 min, with a mound of NPs settling at the bottom after 10 min. The solution became clear after 20 min, and the NPs were completely aggregated at the bottom of the container under the influence of the external magnetic field. This demonstrates that the modified NPs retain a high sensitivity to external magnetic fields and can be efficiently recovered. Figure 11b illustrates that foam stability was minimally affected during the first three recovery cycles, with a reduction in drainage half-life of only 2–7%. However, after the fourth recovery cycle, the drainage half-life decreased significantly, negatively impacting foam stability, with a 19% reduction observed by the fifth cycle. Therefore, to maintain optimal foam stability, the NPs should be recovered and reused no more than three times.

### 3.4. Microscopic Oil Displacement Mechanism of Modified NPs

Water, AOS foam (0.2 wt%), and Fe_3_O_4_@SiO_2_ (0.1 wt%)-AOS (0.2 wt%) foam with a contact angle of 77.01° were employed for oil displacement experiments. The displacement process of specific types of residual oil within a fixed area was monitored in real-time using microscopy. Figure 12 illustrates the effects of the three displacement agents on clustered, membranous, and dead-end residual oil. Due to the high water–oil mobility ratio, water flooding tends to bypass oil, causing water channeling through larger pores and leaving oil trapped in smaller pores, resulting in the formation of clustered residual oil, as depicted in Figure 12a. Membranous residual oil is predominantly distributed in water-flooded regions, where oil adheres to pore surfaces and is not fully displaced by water or surfactant solutions, as shown in Figure 12b. Dead-end residual oil is commonly found in rock formations, characterized by pores connected to the pore-throat network at only one end. Owing to the low viscosity of water, it cannot penetrate dead-end pores, leading to the retention of dead-end residual oil, as illustrated in Figure 12c.

Compared with water flooding, AOS foam demonstrates a stronger capability to mobilize these three types of residual oil. However, Fe_3_O_4_@SiO_2_ (0.1 wt%)-AOS (0.2 wt%) foam exhibits an even more effective displacement performance for these residual oils. As illustrated in the injection pressure curve (Figure 13), the pressure generated during the injection of AOS foam (0.2 wt%) is significantly higher than that of water injection. Furthermore, the injection pressure of Fe_3_O_4_@SiO_2_ (0.1 wt%)-AOS (0.2 wt%) foam is markedly greater than that of AOS foam alone. This indicates that Fe_3_O_4_@SiO_2_ (0.1 wt%)-AOS (0.2 wt%) foam can more effectively perform plugging and profile control functions, thereby enhancing the displacement efficiency of the injected fluid. The enhanced performance is attributed to the synergistic effects of the NPs and foam, which improve the stability and viscoelasticity of the displacing fluid, thereby enhancing its ability to access and displace oil from complex pore structures.

To investigate the mechanism by which Fe_3_O_4_@SiO_2_-(0.1 wt%)-AOS (0.2 wt%) foam with a contact angle of 77.01° mobilizes the three types of residual oil, typical images of the displacement process are shown in Figure 14. As shown in Figure 14a, the foam stabilized by modified Fe_3_O_4_@SiO_2_ NPs blocked the water channeling paths around the clustered residual oil, allowing bubbles to enter low-permeability pores, emulsifying and dispersing the clustered residual oil into membranous oil or small oil droplets, which were then displaced. As shown in Figure 14b, the mobilization of membranous residual oil by modified Fe_3_O_4_@SiO_2_ NP-stabilized foam occurred in two stages: first, the foam emulsified and peeled off the oil attached to the pore surfaces, reducing the difficulty of displacement; second, due to the adsorption of NPs, foam stability was improved, reducing gas diffusion between bubbles and the surface area of the dispersed residual oil, allowing the foam to continuously regenerate and propagate in narrow pores, pushing the oil droplets out. As shown in Figure 14c, the foam stabilized by Fe_3_O_4_@SiO_2_-AOS had good stability, maintaining relatively small bubbles for a longer time, reducing the flow resistance when entering dead-end pores, and allowing deeper penetration into oil-containing dead-end pores. In the dead-end pores, the bubbles deformed under the driving force due to the higher interfacial viscoelasticity of Fe_3_O_4_@SiO_2_-AOS bubbles. The deformed bubbles tended to restore their original shape, gradually replacing the residual oil with films or droplets, and effectively displacing oil from the dead-end pores.

The residual oil in the micromodel was recorded using a microscope, and the images were processed using Image-J software, as shown in Figure 15, to quantitatively calculate the oil recovery rate. Figure 15a,e show the initial oil-saturated state of the micromodel, while Figure 15b–d,f–h show the residual oil after water flooding, foam flooding, and modified NP-stabilized foam flooding, respectively. Compared with the water flooding recovery rate of 42.07%, foam flooding effectively improved the sweep efficiency, compensating for the shortcomings of water flooding and increasing the recovery rate by approximately 16%. However, significant residual oil remained after foam flooding. In contrast, the modified NP-stabilized foam flooding had a larger sweep efficiency and higher displacement efficiency, achieving a recovery rate of 75.40%, an increase of 33.33% compared with water flooding.

## 4. Conclusions

Through shell coating and hydrophobic modification experiments on Fe_3_O_4_ NPs, Fe_3_O_4_@SiO_2_ NPs with different wettabilities were prepared, and their foam-stabilization performance was evaluated. The effects of modified NPs on foam stability were assessed through static foam stability experiments, including drainage half-life and initial foam volume. The foam-stabilization mechanism of Fe_3_O_4_@SiO_2_ NPs was investigated using confocal laser scanning microscopy. Microscopic oil displacement experiments were conducted to evaluate the enhanced oil recovery mechanism of modified NP-stabilized foam.

The main findings are as follows:(1)The characterization of the prepared NPs and foam stability evaluation showed that Fe_3_O_4_@SiO_2_-1.0 NPs with a contact angle of 77.01° had the best foam-stabilization performance, significantly improving foam stability. The optimal foam system consisted of 1 wt% NPs (77.01°) + 0.2 wt% SDS, with a drainage half-life of 452 s and an initial foam volume of 510 mL.(2)Confocal laser scanning microscopy experiments showed that the modified Fe_3_O_4_@SiO_2_ NPs were adsorbed on the bubble surface, forming a three-dimensional network structure between armored bubbles, thereby enhancing foam stability. A static foam stability evaluation indicated that the optimal number of NP recovery cycles was three, and Fe_3_O_4_@SiO_2_ NPs responded quickly to magnetic fields.(3)Microscopic visual oil displacement experiments demonstrated that, compared with AOS foam alone, Fe_3_O_4_@SiO_2_-1000 NP-stabilized foam had a higher ability to mobilize residual oil. The foam’s strong stability blocked large pores, allowing subsequent fluids to enter small pores, emulsifying and mobilizing clustered residual oil. The dense adsorption of modified Fe_3_O_4_ NPs at the liquid film interface significantly enhances film strength, enabling bubbles to undergo elastic deformation rather than rupture when passing through pore throats, emulsifying and peeling off membranous residual oil and pushing it out. Bubbles entered dead-end pores through high viscoelastic deformation, carrying out residual oil. Compared with the water flooding recovery rate of 42.07%, the modified NP-stabilized foam achieved a recovery rate of 75.40%, an increase of 33.33%, effectively mobilizing residual oil.

## Figures and Tables

**Figure 1 nanomaterials-15-00395-f001:**
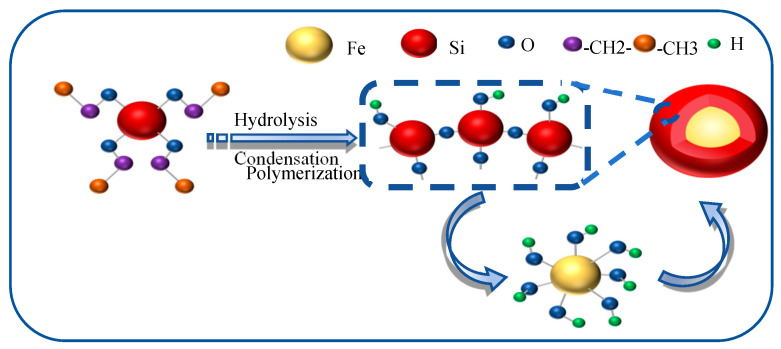
Schematic representation of the synthesis mechanism of Fe_3_O_4_@SiO_2_ NPs.

**Figure 2 nanomaterials-15-00395-f002:**
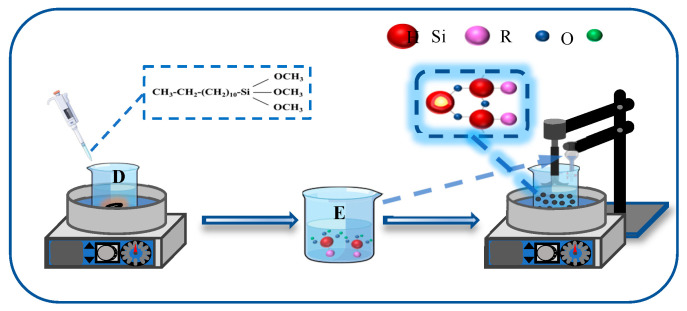
Synthesis process of modified Fe_3_O_4_@SiO_2_ NPs.

**Figure 3 nanomaterials-15-00395-f003:**
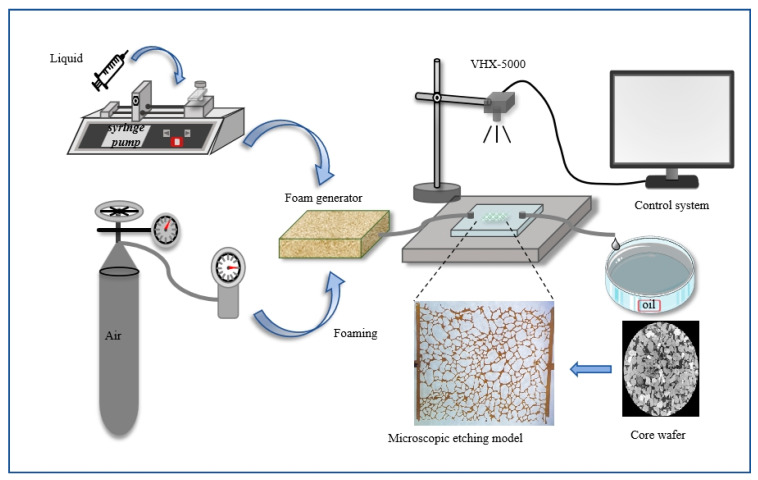
Flow chart of microscopic oil displacement experiment.

**Figure 4 nanomaterials-15-00395-f004:**
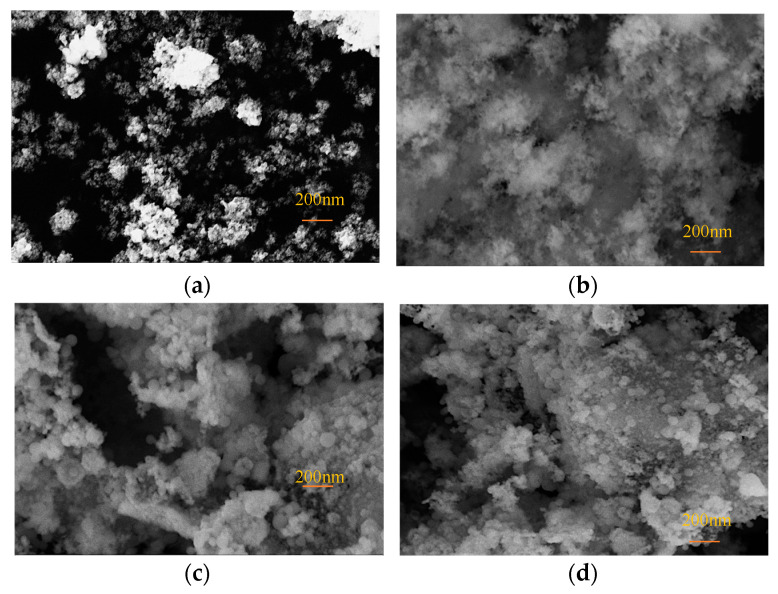
SEM images of NPs: (**a**) Fe_3_O_4_; (**b**) Fe_3_O_4_@SiO_2_-800; (**c**) Fe_3_O_4_@SiO_2_-1000; (**d**) Fe_3_O_4_@SiO_2_-1200.

**Figure 5 nanomaterials-15-00395-f005:**
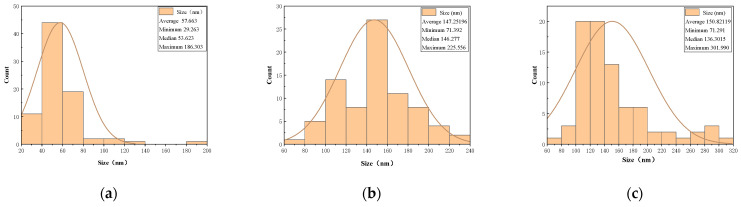
Size distribution of NPs: (**a**) Naked Fe_3_O_4_; (**b**) Fe_3_O_4_@SiO_2_-1000; (**c**) Fe_3_O_4_@SiO_2_-1200.

**Figure 6 nanomaterials-15-00395-f006:**
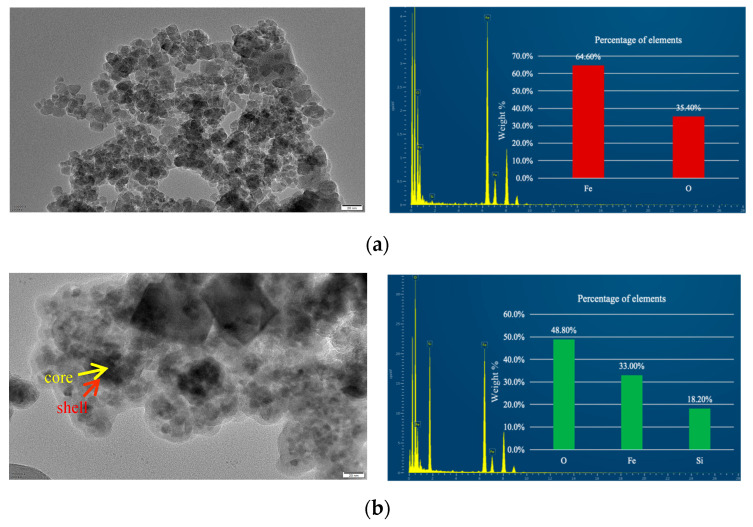
TEM images and EDS of nanoparticles (**a**) Fe_3_O_4_; (**b**) Fe_3_O_4_@SiO_2_-1000.

**Figure 7 nanomaterials-15-00395-f007:**
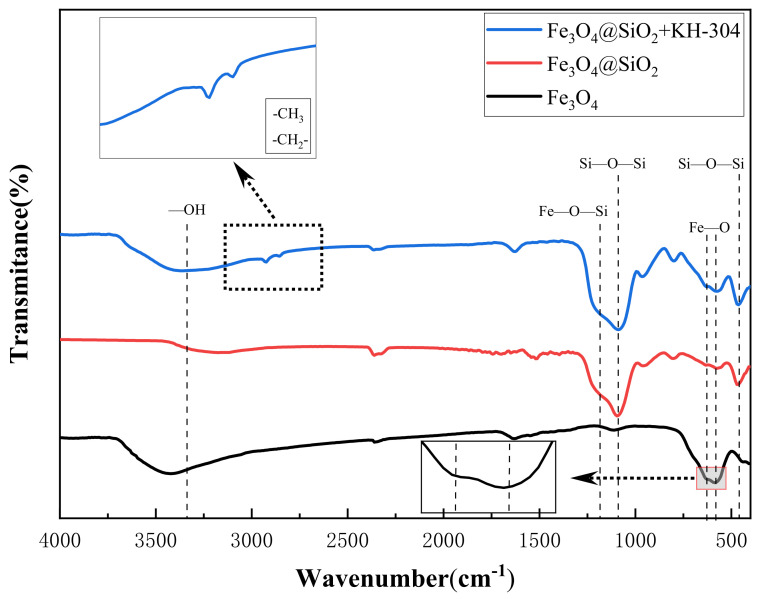
Infrared spectrum of NPs.

**Figure 8 nanomaterials-15-00395-f008:**
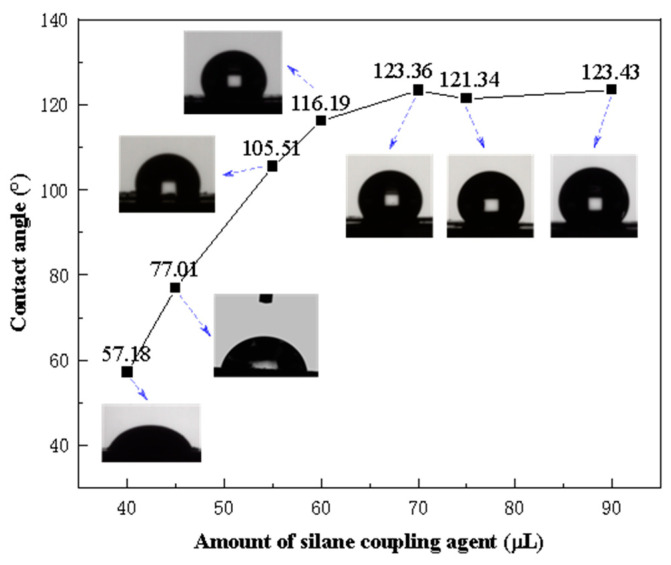
The contact angles of NPs with water as a function of amounts of silane coupling agent.

**Figure 9 nanomaterials-15-00395-f009:**
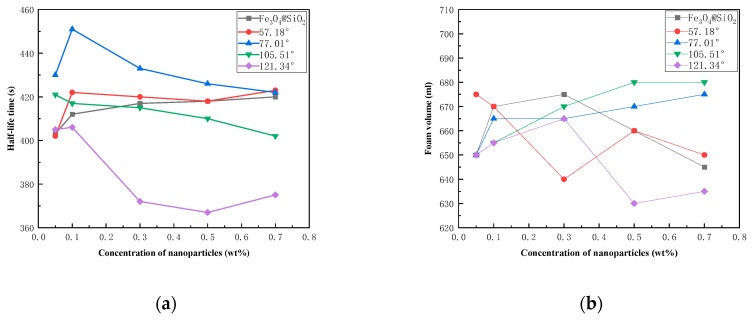
The different wettability nanoparticles on stabilizing foam: (**a**) drainage half-life time; (**b**) foam volume.

**Figure 10 nanomaterials-15-00395-f010:**
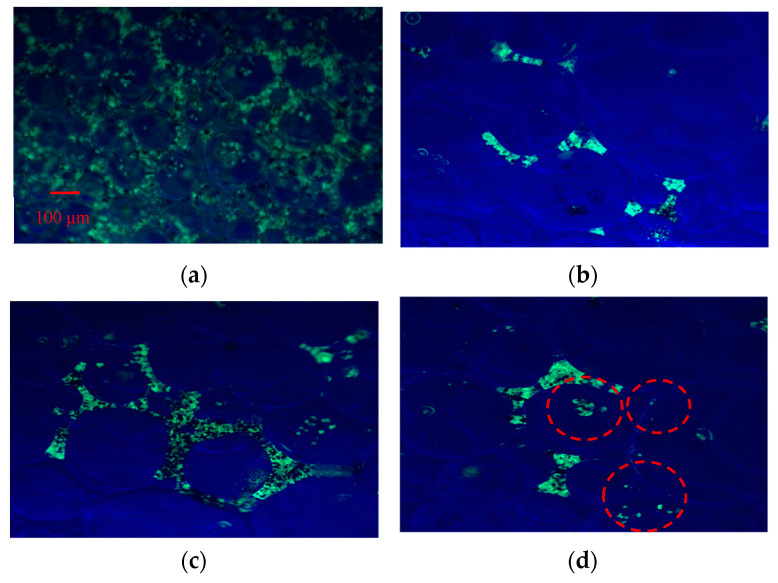
Confocal fluorescence image for the foams (Fe_3_O_4_@SiO_2_ NPs-AOS (0.2wt%)). (**a**) Wet foam; (**b**) dry foam; (**c**) dry foam stabilized by NPs with a contact angle of 77.01; (**d**) dry foam stabilized by NPs with a contact angle of 121.34°. The red circles are bubbles.

**Figure 11 nanomaterials-15-00395-f011:**
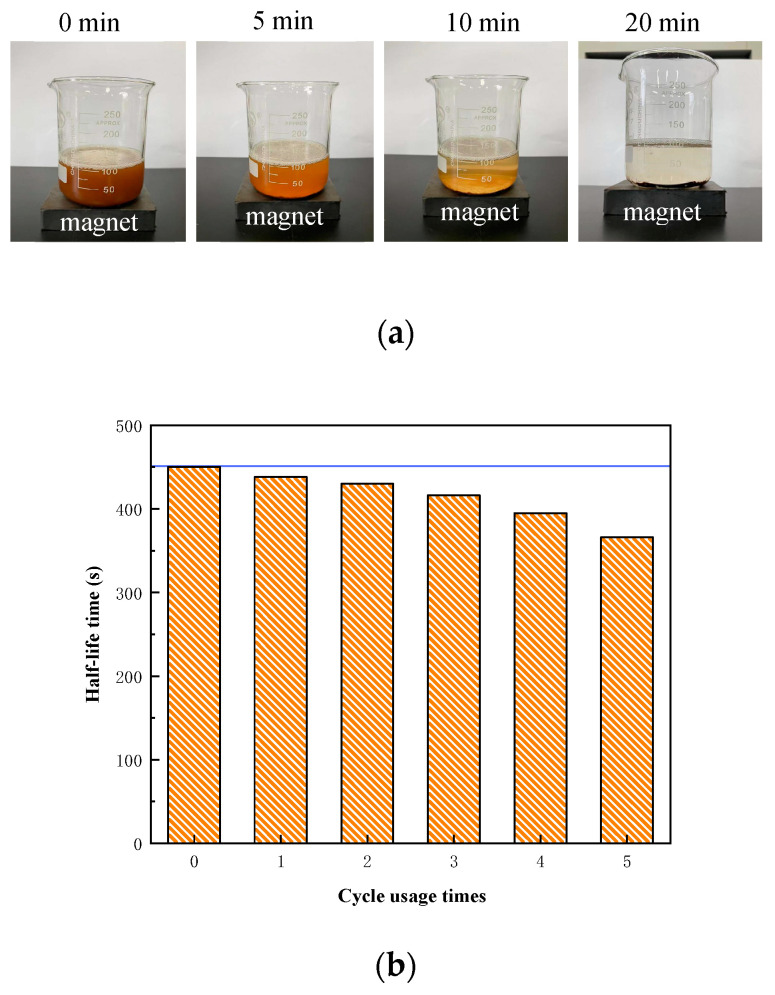
Evaluation of recyclability of NPs: (**a**) the responsiveness of NPs to a magnet; (**b**) the drainage half-life time changes with the recycle number of NPs.

**Figure 12 nanomaterials-15-00395-f012:**
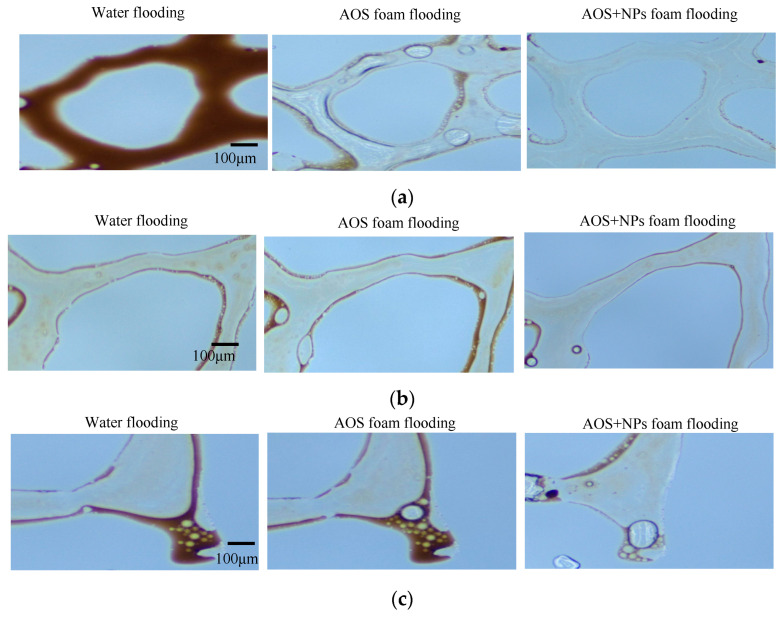
Microscopic residual oil after flooding: (**a**) cluster residual oil; (**b**) membranous residual oil; (**c**) dead-end residual oil.

**Figure 13 nanomaterials-15-00395-f013:**
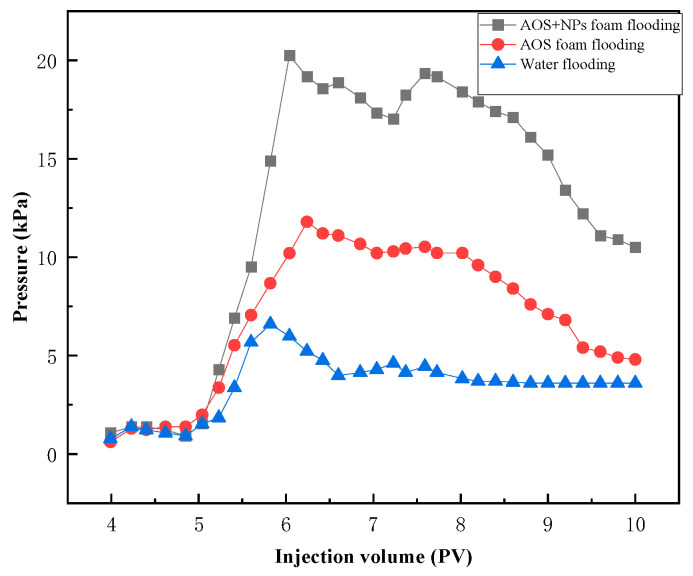
The pressure of flooding experiments.

**Figure 14 nanomaterials-15-00395-f014:**
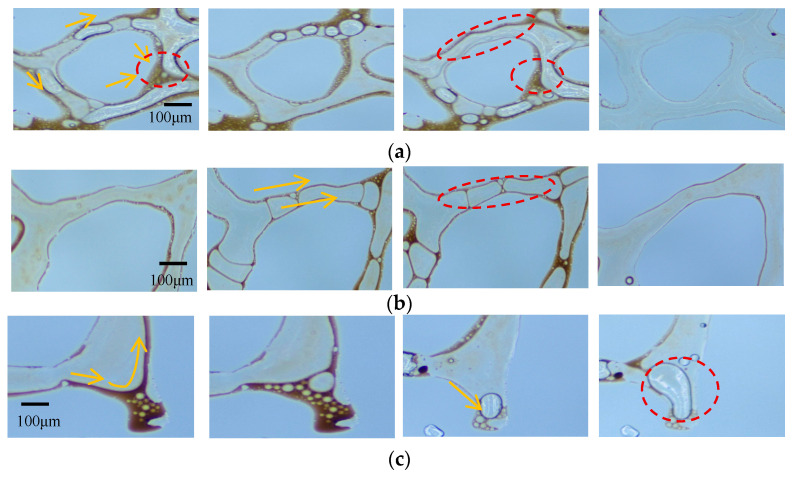
Microscopic residual oil after Fe_3_O_4_@SiO_2_-AOS foam flooding: (**a**) cluster residual oil; (**b**) membranous residual oil; (**c**) dead-end residual oil.

**Figure 15 nanomaterials-15-00395-f015:**
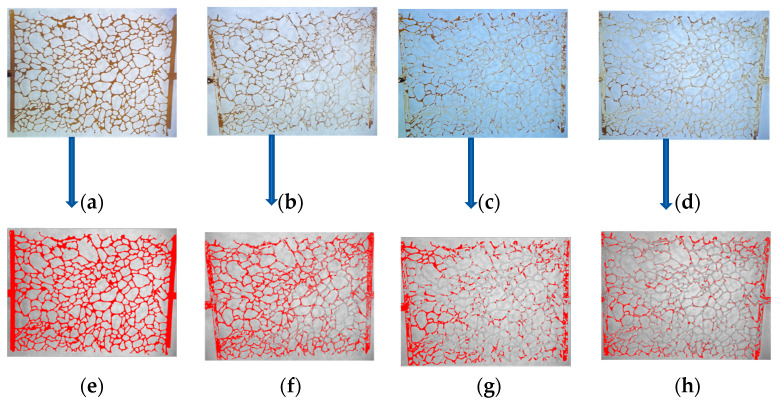
Microscopic images of residual oil distributions in 2D micromodel. (**a**) The raw images of initial oil distribution, (**b**) residual oil distributions after water flooding, (**c**) AOS foam flooding and extended water flooding, and (**d**) Fe_3_O_4_@SiO_2_-AOS foam flooding and extended water flooding, respectively, while (**e**–**h**) were processed versions of images (**a**–**d**) using Image J software to estimate trapped oil. The dark brown color is oil and the milky color is displacement fluid in (**a**–**d**) whereas red is oil and the ash color is displacement fluid in (**e**–**h**). The flow direction is from left to right.

## Data Availability

The authors confirm that the data supporting the findings of this study are available within the article.

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
