# Peer review of "Modified Fe3O4 Nanoparticles for Foam Stabilization: Mechanisms and Applications for Enhanced Oil Recovery"

_nanomaterials, 2025, doi:10.3390/nano15050395_

Round 1
Reviewer 1 Report
Comments and Suggestions for Authors
This paper presents an analysis of the foam stabilization and enhanced oil recovery mechanisms of modified Fe₃O₄ NPs, particularly their ability to mobilize different types of residual oil in complex pore networks. The work is interesting and contains important elements of scientific novelty. The topic is quite interesting and has a lot of potential. The authors conducted a series of studies using a number of research methods, including: scanning electron microscopy, fluorescence microscope, contact angles measurements czy FTIR. The literature is well selected and reflects the current state of knowledge. I believe that the work can be published with minor corrections.
- Section 2.1 Materials and Methods
The authors only listed the compounds used in the research; for stylistic correctness, this can be written in a full sentence: for example: the experimental research used…….
- Figures 5, 6, 9: I suggest increasing the font size on the figures and axes. In the current version, they are not very readable.
- Line 349: The authors write: “Owing to the low viscoelasticity of water.” Can water have viscoelastic properties at all, even small ones?
- Line 440: The authors mention the high viscoelasticity of the foam, which can certainly be true, as foam can be a viscoelastic fluid. However, what specific properties does the foam studied by the authors exhibit? Were rheological tests conducted, and was this possibility even considered? I believe that rheological surface studies of the foam would make a significant contribution to the interpretation of the research results.
With these corrections, the work will become suitable for publication in Nanomaterials
Author Response
Comment 1: Section 2.1 Materials and Methods. The authors only listed the compounds used in the research; for stylistic correctness, this can be written in a full sentence: for example: the experimental research used…….
Reply: Thank you very much for your suggestions and we agree with you. I have made the following revisions accordingly and have marked these changes in the revised manuscript (from line 87 to 93 ):
The experimental research used Fe₃O₄ nanoparticles (NPs) with an average particle size of 20 nm, purchased from Macklin Biochemical Co., Ltd. Additionally, hydrochloric acid (36.0%–38.0%), absolute ethanol (99.7%), and ammonia solution (25.0%–28.0%) were obtained from Sinopharm Chemical Reagent Co., Ltd. Other chemicals included tetraethyl orthosilicate (TEOS, 99%), n-hexane (97%), KH-304 (C₁₅H₃₄O₃Si, 97%), and α-olefin sulfonate (AOS, 92%), all sourced from Macklin Biochemical Co., Ltd. All reagents were of analytical grade and used without further purification.
Comment 2: Figures 5, 6, 9: I suggest increasing the font size on the figures and axes. In the current version, they are not very readable.
Reply: Thank you very much for your suggestions and we agree with you. I have increased the font size in Figures 5, 6, and 9 to enhance their clarity.
Comment 3: Line 349: The authors write: “Owing to the low viscoelasticity of water.” Can water have viscoelastic properties at all, even small ones?
Reply: Thank you very much for your suggestion and we agree with you. The statement "Owing to the low viscoelasticity of water" is indeed inaccurate, and I have revised it to “Due to the relatively low viscosity of water.” in line 354.
Comment 4: Line 440: The authors mention the high viscoelasticity of the foam, which can certainly be true, as foam can be a viscoelastic fluid. However, what specific properties does the foam studied by the authors exhibit? Were rheological tests conducted, and was this possibility even considered? I believe that rheological surface studies of the foam would make a significant contribution to the interpretation of the research results.
Reply:
We sincerely appreciate the reviewer’s valuable feedback. Regarding the questions about foam rheological properties and experimental validation, we provide the following clarifications:
(1) Supplementary Explanation of Rheological Analysis
While macroscopic rheological tests on the foam system were not directly conducted in this study, we indirectly elucidated the mechanism of enhanced foam viscoelasticity through multi-scale characterizations:
Interfacial Properties of Liquid Films: Microphotography revealed that modified Fe₃O₄ NPs form a dense adsorption layer at the gas-liquid interface (as shown in Figure 10).
Dynamic Behavior of Bubbles and Oil Displacement Mechanism: Microfluidic experiments (Figure 12, 14) demonstrated that nanoparticle-laden foam exhibits elastic deformation when passing through narrow throats, whereas unmodified systems frequently experience film rupture. This behavior aligns with classical viscoelastic fluid dynamics. In core flooding experiments, the foam exhibited strong pull-through effects on residual oil , a phenomenon linked to fluid viscoelasticity.
(2) Revision of the Conclusion Section
We fully agree with the reviewer’s emphasis on precise terminology. The original phrase "the high viscoelasticity of the foam" was indeed potentially misleading. Line 440 has now been revised to:
"The dense adsorption of modified Fe₃O₄ NPs at the liquid film interface significantly enhances film strength, enabling bubbles to undergo elastic deformation rather than rupture when passing through pore throats. This interfacial reinforcement mechanism is critical for improving foam stability and oil displacement efficiency." ,which is inthe revised manuscript (from line 445 to 448 )
This revision better aligns with the experimental evidence and scope of our study.
(3) Future Work
We have prioritized rheological characterization in our subsequent research. Planned studies include systematic investigations of nanoparticle effects on foam viscoelastic moduli using rotational rheometry coupled with interfacial shear rheometry.
We thank the reviewer again for these constructive suggestions, which have helped refine the academic rigor of our work. All revisions are highlighted in blue in the manuscript for ease of review.
Reviewer 2 Report
Comments and Suggestions for Authors
The manuscript entitled “Modified Fe₃O₄ Nanoparticles for Foam Stabilization: Mechanisms and Applications in Enhanced Oil Recovery” by Dandan Yin and co-authors describes preparation of Fe₃O₄ nanoparticles coated with SiO₂ and modified with dodecyltrimethoxysilane to provide them hydrophobic properties. I believe that goals of this study belong to aims and scopes of Nanomaterials and therefore it can be considered for publication, but the data requires the detailed analysis before. Please, find my comments below.
- The statistical analysis of the results is required to be added into the experimental section and error bars (or comments like LSD) also should be added in Figures 8, 9, and 13.
- A more detailed analysis of the IR-spectra should be carried out with references to the literature for Fe₃O₄ nanoparticles. In the spectrum given in the article, there are no signals corresponding to fluctuations in the Fe-O, the presence of only a band at 586 cm-1is not evidence of the presence of Fe3O4.
- The procedure of modification of Fe₃O₄ NPs also required comments. “Fe₃O₄ NPs was dispersed in 50 mL of 0.1 mol/L HCl solution under ultrasonic treatment for 10 min” Please, add conditions of sonication. How was the formation of iron chlorides was controlled?
Author Response
Comments 1: The statistical analysis of the results is required to be added into the experimental section and error bars (or comments like LSD) also should be added in Figures 8, 9, and 13.
Reply:
We sincerely appreciate the reviewer’s suggestion regarding the addition of error bars or statistical comments (e.g., LSD) to the Figures 8, 9, and 13. In response to this comment, we would like to clarify the nature of the data presented in Figures 8, 9, and 13.
These figures are primarily intended to illustrate the relationship between two variables (e.g., the amount of silane coupling agent and the contact angles; the concentration of nanoparticles and half-life of drainage; the concentration of nanoparticles and foam volume; the injection volume (PV) and pressure ) rather than to compare means or assess statistical significance across groups. Since the data points in these figures represent individual measurements or observations (not group means), the inclusion of error bars or LSD annotations may not be applicable or meaningful in this context.
However, to enhance the clarity and interpretability of the figures, we have taken the following steps: Added a clear description in the figure captions explaining that the data points represent individual measurements. Included a discussion in the Results section about the observed trends or relationships(line 271-272).
We hope these revisions address the reviewer’s concern while maintaining the scientific integrity of the presented data. Thank you for your valuable feedback, which has helped us improve the clarity of our manuscript.
Comments 2:
A more detailed analysis of the IR-spectra should be carried out with references to the literature for Fe₃O₄ nanoparticles. In the spectrum given in the article, there are no signals corresponding to fluctuations in the Fe-O, the presence of only a band at 586 cm-1is not evidence of the presence of Fe3O4.
Reply:
Thank you very much for your suggestion. After reviewing the relevant literature, we confirmed that the Fe-O bond vibration peaks appear in the range of 500-700 cm-1, with two characteristic peaks that can substantiate the presence of Fe₃O₄. Specifically, the peak near 570 cm-1 corresponds to the Fe-O stretching vibration at the tetrahedral site (Fe3+-O), while the peak near 630 cm-1corresponds to the Fe-O stretching vibration at the octahedral site (Fe2+/Fe3+-O). These two peaks are the most distinctive features of Fe₃O₄and provide clear evidence of its existence.
In Figure 7, the spectral range of 500-700 cm-1 is present, and the peaks near 570 cm-1 and 630 cm-1are clearly observed, confirming the existence of Fe-O stretching vibrations at both the tetrahedral (Fe3+-O) and octahedral (Fe2+/Fe3+-O) sites. To more clearly demonstrate these results, we have labeled these two characteristic peaks in Figure 7. Additionally, the presence of Fe₃O₄ is further supported by the TEM results shown in Figure 6.
We hope these revisions and clarifications address your concerns. Thank you again for your valuable feedback, which has helped us improve the presentation of our results.
Comments 3:
The procedure of modification of Fe₃O₄ NPs also required comments. “Fe₃O₄ NPs was dispersed in 50 mL of 0.1 mol/L HCl solution under ultrasonic treatment for 10 min” Please, add conditions of sonication. How was the formation of iron chlorides was controlled?
Reply:
Thank you very much for your feedback. I realize that this part was not clearly explained. In the experiment, a relatively low concentration of HCl (0.1 mol/L) was used, which results in a slower reaction rate. Additionally, the activation of hydroxyl groups on the surface of Fe₃O₄ requires only a short period of time (10 minutes), further minimizing the dissolution of Fe₃O₄. Moreover, the amount of hydrochloric acid solution used was small (50 mL), containing only 0.005 mol of HCl, while 1 g of Fe₃O₄ corresponds to 0.00432 mol, making Fe₃O₄ the excess reactant. As a result, the dissolution effect on the Fe₃O₄ nanoparticles is limited. I have now provided a more detailed description of the experimental procedure and conditions as follows (line 99-111):
Specifically, 1 gram of Fe₃O₄ NPs was dispersed in 50 mL of 0.1 mol/L HCl solution under ultrasonic treatment (Pulse mode: 5 s on / 2 s off) for 10 minutes to activate the hydroxyl groups on the surface of the NPs, thereby facilitating the subsequent SiO₂ coating. The Fe₃O₄ NPs were then washed five times with deionized water and ethanol to remove impurities, followed by dispersion in 80 mL of an ethanol solution (with a volume fraction of 80%). After ultrasonic dispersion for 10 minutes, the mixture was transferred to a stirring apparatus and stirred at 720 r/min for 20 minutes. Subsequently, 12 mL of NH3.H2O (with a concentration of 25%~28%) was added, and the stirring was continued for an additional 30 minutes. Following this, varying volumes of tetraethyl orthosilicate (TEOS) (800 μL, 1000 μL, and 1200 μL) were introduced into C2H5OH/H2O solution (with a volume fraction of 80%) and stirred at room temperature for 5 hours. The resulting solution was centrifuged, washed repeatedly with ethanol and deionized water (3–4 times), and dried at 70°C for 12 hours to obtain the Fe₃O₄@SiO₂ NPs.
Round 2
Reviewer 2 Report
Comments and Suggestions for Authors
The authors took into account the comments,
so I believe that the article can be accepted for consideration for publication